# Structural and Mechanistic Basis for the Inactivation of Human Ornithine Aminotransferase by (3*S*,4*S*)-3-Amino-4-fluorocyclopentenecarboxylic Acid

**DOI:** 10.3390/molecules28031133

**Published:** 2023-01-23

**Authors:** Sida Shen, Arseniy Butrin, Brett A. Beaupre, Glaucio M. Ferreira, Peter F. Doubleday, Daniel H. Grass, Wei Zhu, Neil L. Kelleher, Graham R. Moran, Dali Liu, Richard B. Silverman

**Affiliations:** 1Department of Chemistry and Center for Developmental Therapeutics, Northwestern University, Evanston, IL 60208, USA; 2Department of Chemistry and Biochemistry, Loyola University Chicago, Chicago, IL 60660, USA; 3Department of Clinical and Toxicological Analyses, School of Pharmaceutical Sciences, University of São Paulo, São Paulo 05508-000, SP, Brazil; 4Department of Molecular Biosciences, Northwestern University, Evanston, IL 60208, USA; 5Proteomics Center of Excellence, Northwestern University, Evanston, IL 60208, USA; 6Department of Pharmacology, Northwestern University, Chicago, IL 60611, USA

**Keywords:** aminotransferase, mechanism-based inactivators, enamine, cyclopentene, mechanistic studies

## Abstract

Ornithine aminotransferase (OAT) is overexpressed in hepatocellular carcinoma (HCC), and we previously showed that inactivation of OAT inhibits the growth of HCC. Recently, we found that (3*S*,4*S*)-3-amino-4-fluorocyclopentenecarboxylic acid (**5**) was a potent inactivator of γ-aminobutyric acid aminotransferase (GABA-AT), proceeding by an enamine mechanism. Here we describe our investigations into the activity and mechanism of **5** as an inactivator of human OAT. We have found that **5** exhibits 10-fold less inactivation efficiency (*k*_inact_/*K*_I_) against *h*OAT than GABA-AT. A comprehensive mechanistic study was carried out to understand its inactivation mechanism with *h*OAT. p*K*_a_ and electrostatic potential calculations were performed to further support the notion that the α,β-unsaturated alkene of **5** is critical for enhancing acidity and nucleophilicity of the corresponding intermediates and ultimately responsible for the improved inactivation efficiency of **5** over the corresponding saturated analogue (**4**). Intact protein mass spectrometry and the crystal structure complex with *h*OAT provide evidence to conclude that **5** mainly inactivates *h*OAT through noncovalent interactions, and that, unlike with GABA-AT, covalent binding with *h*OAT is a minor component of the total inhibition which is unique relative to other monofluoro-substituted derivatives. Furthermore, based on the results of transient-state measurements and free energy calculations, it is suggested that the α,β-unsaturated carboxylate group of PLP-bound **5** may be directly involved in the inactivation cascade by forming an enolate intermediate. Overall, compound **5** exhibits unusual structural conversions which are catalyzed by specific residues within *h*OAT, ultimately leading to an enamine mechanism-based inactivation of *h*OAT through noncovalent interactions and covalent modification.

## 1. Introduction

Aminotransferases are members of the pyridoxal 5′-phosphate (PLP)-dependent class of enzymes. They are capable of performing two coupled transamination reactions between an amino acid and an α-keto acid, participating in essential nitrogen metabolism in vivo [1]. The PLP moiety links to a catalytic lysine (Lys) residue through a Schiff base, forming an internal PLP–Lys aldimine structure, which can specifically react with an amino acid substrate to produce a keto analog while forming pyridoxamine 5′-phosphate (PMP), thereby completing the first transamination reaction. The PMP then forms another Schiff base with an α-keto acid substrate during the second transamination reaction catalyzed by the aminotransferase to generate an α-amino acid and the original internal PLP–Lys aldimine cofactor [2].

Several aminotransferases have been recognized as therapeutic targets for neurological disorders and cancers. For example, γ-aminobutyric acid aminotransferase (GABA-AT, E.C. 2.6.1.19) functions to degrade the primary inhibitory neurotransmitter GABA to succinic semialdehyde (SSA) (Figure 1A) while producing the major excitatory neurotransmitter L-glutamate (L-Glu) from α-ketoglutarate (α-KG) [3]. The inhibition of GABA-AT has emerged as a therapeutic approach for epilepsy which acts by balancing the reduced GABA levels detected in these patients’ central nervous systems [4]. Human ornithine δ-aminotransferase (*h*OAT; EC2.6.1.13), which belongs to the same evolutionary subgroup as GABA-AT, converts L-ornithine (L-Orn) into L-glutamate-γ-semialdehyde (L-GSA) while also producing L-Glu from α-KG (Figure 1B) [5]. We previously found that the *OAT* gene was overexpressed in spontaneous hepatocellular carcinoma (HCC)-developing livers from sand rats [6]. Moreover, pharmacological inhibition of *h*OAT significantly inhibited tumor growth in an HCC mouse model [6]. A recent study also revealed that *h*OAT knockdown suppressed the tumor growth in a non-small cell lung cancer (NSCLC) mouse model [7]. Consequently, *h*OAT has emerged as a potential target for developing effective treatments for different types of cancers; therefore, we are interested in determining the mechanistic differences among compounds that inactivate both GABA-AT and *h*OAT for the future design of selective inactivators of either GABA-AT or *h*OAT as potential therapeutics.

To date, vigabatrin (VGA, γ-vinyl-GABA, Figure 1) is the only FDA-approved GABA-AT inhibitor for the treatment of infantile spasms that acts as a mechanism-based inactivator (MBI) [3,8]. MBIs initially react in a similar manner to substrates in the active site of the targeted enzymes and are converted into an activated species, which usually leads to inactivation either by covalent bond formation or tight-binding inhibition [9]. Mechanistic and crystallographic studies have demonstrated that 70% of vigabatrin inactivation of GABA-AT occurs through a Michael addition pathway: after deprotonation (**M1** to **M2**) and tautomerization (**M2** to **M3**) steps, electrophile **M3** reacts with the catalytic Lys329 residue in GABA-AT to form a covalent bond (**M4** and **M5**). Thirty percent of vigabatrin’s inactivation occurs via tautomerization of the vinyl group (**M2** to **M6**); an attack by Lys329 generates enamine intermediate **M7**. This reacts with the internal PLP–Lys329 aldimine, leading to inactivated complex **M8** (Figure 1) [10].

Inspired by the inactivation mechanism of vigabatrin, (*S*)-4-amino-5-fluoropentanoic acid **1**, AFPA) was initially designed as a Michael addition mechanism-based GABA-AT inactivator [11]. After deprotonation (**M9** to **M10**) and fluoride ion elimination (**M10** to **M11**), it was anticipated that electrophile **M11** would form **M12** then **M13** from nucleophilic attack by an active site residue (Pathway a; Figure 2). However, subsequent mechanistic studies revealed that the Lys329 residue reacts exclusively with the imine linkage of **M11**, releasing enamine intermediate **M14** and forming covalently bound **M15** (Pathway b; Figure 2) [12]. Therefore, AFPA inactivates GABA-AT through an enamine addition mechanism rather than via the initially proposed Michael addition. Since this discovery, a number of alkyl chain and cyclized analogs bearing a monofluoro group have been demonstrated to be enamine addition MBIs of GABA-AT or *h*OAT (Table 1) [13,14,15,16,17]. Because of the structural similarity between GABA-AT and *h*OAT, cyclopentane-based FCP (**4**) was found to exhibit dual inactivation of these two aminotransferases via the same enamine inactivation mechanism (Figure 3) [18]. Similar to the enamine addition pathway of AFPA (Pathway b; Figure 2), the inactivation mechanism of FCP is initiated by the formation of Schiff base **M16**, followed by deprotonation (**M16** to **M17**) and fluoride ion elimination (**M17** to **M18**) to afford intermediate **M18**. The released enamine intermediate **M19** from **M18** reacts with the internal PLP–Lys aldimine to yield the covalent adduct **M20**, whose structure was verified by crystallography and intact protein mass spectrometry (intact protein MS) [13,18,19].

By incorporation of a double bond into the C_α_ and C_β_ positions of FCP (**4**) (Figure 3), we recently found that cyclopentene derivative **5** showed 23-fold improved inactivation efficiency (*k*_inact_/*K*_I_) against GABA-AT (Table 1) [13]. When **5** was tested against *h*OAT, a 9.5-fold increase in inactivation efficiency was observed relative to **4**. This indicates that **5** has the potential for multiple therapeutic activities by the inactivation of both GABA-AT and *h*OAT. In this paper, we synthesized and evaluated another cyclopentene derivative (**6**) bearing an α,ε-conjugated carboxylate. In addition, we utilized computational calculations to elucidate the effects of slight structural differences among **4**–**6** on their inactivation steps, which eventually influence their inhibitory activities. We also carried out mechanistic studies of **5** with *h*OAT using intact MS, transient-state kinetics, and co-crystallization to reveal that compound **5** primarily inactivates *h*OAT through an unusual noncovalent interaction.

## 2. Results and Discussion

### 2.1. Synthesis of Monofluorinated Cyclopentene Analogs 5 and 6

In the 10-step synthetic route that we previously reported for the preparation of **5** [13], an acid-assisted epoxide ring-opening reaction was utilized to form a bicyclic key intermediate containing a hydroxyl group on the bridgehead. The hydroxyl group was tosylated and employed as a leaving group to afford the desired cyclopentene ring through a one-pot lactam hydrolysis and β-elimination. To better support our mechanistic studies, in this work, we first optimized the preparation of **5** to a 6-step synthetic route (Figure 4). Bicyclic lactam intermediate **12**, bearing a bromo group on the bridgehead, was prepared from a chirally pure Vince lactam (**(-)-10**; (1*R*)-(−)-2-azabicyclo[2.2.1]hept-5-en-3-one; CAS#: 79200-56-9) following the procedure reported previously [22]. The hydroxyl-bearing carbon of **12** was fluorinated using DAST to generate **13** as a single diastereomer [13]. After switching the protecting group of the lactam from a PMB group to a Boc group (**14**), the lactam was hydrolyzed with K_2_CO_3_/MeOH. Simultaneously, the bromo group on the bridgehead of **14** was applied as a leaving group to induce β-elimination, forming the desired cyclopentene ring with the α,β-unsaturated carboxylate methyl ester (**15**). The Boc group and methyl ester of **15** were removed with HCl/AcOH to afford the final product (**5**) as an HCl salt.

To prepare monofluorinated cyclopentene analog **6** bearing an α,ε-unsaturated carboxylate (Figure 5), racemic Vince lactam (**(±)-10**; (±)-2-azabicyclo[2.2.1]hept-5-en-3-one; CAS#: 49805-30-3) was first protected with a Boc group and then epoxidized using *m*-CPBA (**17**) [13]. Intermediate **17** underwent a one-pot epoxide ring-opening, elimination, and lactam hydrolysis with K_2_CO_3_/MeOH [23,24], affording cyclopentene **18** an allylic alcohol. The Boc group of **18** was converted into a di-PMB protecting group (**19**) followed by fluorination using Deoxo-Fluor and then PMB/Boc group deprotection and methyl ester hydrolysis to give the final product (**6**) as an HCl salt. The kinetic constants shown in Table 1 indicate that only **6** shows reversible inhibition of GABA-AT (*K*_i_ = 0.14 mM), and no inhibition was observed with *h*OAT up to 10 mM.

### 2.2. The Significance of the Conjugated Alkene of 5

The deprotonation step (e.g., **M16** to **M17** in Figure 3) is considered to be the rate-limiting step for enamine mechanism-based GABA-AT inactivators [3]. Previously, we used quantum mechanical cluster calculations on the deprotonation step, catalyzed by Lys329 in GABA-AT, to demonstrate that the ligand-PLP Schiff base **M16-1** (Figure 2A) [13], containing the added double bond, displays about 3 kcal/mol lower transition state (TS) energy compared with intermediate **M16**. In this work, we performed theoretical *p*K_a_ calculations using the DFT/B3LYP level of theory that we previously published [14,25,26] at 298K to predict the acidity at the C_γ_ position of difluoro analogs **4**–**6** (Table 1). The results, shown in Figure 2A, suggest that the hydrogen (highlighted in red) of PLP-bound **5** (**M16-1**) with an α, β-unsaturated carboxylate system displays the lowest *p*K_a_ value among the three analogs, while the *p*K_a_ value of the corresponding proton in PLP-bound **6** (**M16-2**) is not noticeably affected by the introduction of the double bond at the C_α_ and C_ε_ positions relative to parent cyclopentane **4** (**M16**). Electrostatic potential (ESP) calculations [14] (Figure 2B) indicate that the nucleophilicity of C_δ_ in the enamine intermediate of **5** (**M19-1**) is much higher than that in the corresponding enamines of **4** and **6** (**M19** for **4** and **M19-2** for **6**), suggesting that the C_δ_ position of **M19-1** is more reactive than that in the other two intermediates, leading to more efficient enamine addition. In contrast, the enamine intermediate of **6** (**M19-2**) exhibits the lowest nucleophilicity, which should retard the attack at the internal aldimine, resulting in the modest inhibition of **6** on *h*OAT.

### 2.3. The Unusual Covalent and Noncovalent Interactions between hOAT and the Products Generated from 5

Three different enamine mechanism-based inactivators—**4**, **5**, and **8**—were used to fully inactivate *h*OAT, and these samples were utilized in denaturing the intact protein MS [14] to evaluate the covalent inactivation adducts formed for these three compounds. All the covalent mass shifts match the ketone adduct generated from the enamine addition pathway (Figure 3). Intriguingly, the major mass peaks are the modified peaks in the *h*OAT samples inactivated by **4** and **8** (Figure 3, left and right). In contrast, the apo-*h*OAT peak is the primary one in the sample treated with **5**, while only a small portion of *h*OAT was covalently modified by **5** (Figure 3, middle). According to our prior experience [26], these differences indicate that, unlike **4** and **8**, that noncovalent interaction is the predominant inactivation form in the case of **5**.

The subsequent wavelength absorbance measurement for **4**, **5**, and **8** after a 4 h incubation with *h*OAT (Figure 4, right) suggests that the final products of **4** and **8** exhibit maximal absorption at about 330 nm. This is consistent with the theoretical wavelengths (320–330 nm) for the final product generated from the enamine addition (Figure 4, left). In contrast, the final product of **5** displays a distinct maximal absorption at 460 nm.

We further carried out co-crystallization experiments to explore the final product of **5** in *h*OAT. The *h*OAT crystals grew within three days and showed orthorhombic morphology, with the largest dimension of ~0.2 mm. Crystals were transferred to a cryo-protectant solution and flash-frozen in liquid nitrogen. The acquired crystals diffracted to a resolution of 1.91 Å (Appendix A). The crystal structure was solved using molecular replacement. The most well refined molecular model had three monomers in an asymmetric unit in the C121 space group. The overall protein fold of the monomer is a typical representative of a subgroup 2 aminotransferase [27]. Each monomer consists of an N-terminal domain that contains a loop, a helix, and a three-stranded β-meander, a small C-terminal domain that is most distant to the 2-fold axis, and a large domain that contributes most to the interface of the subunit. In an asymmetric unit, the monomers are related to each other by a 3-fold screw axis (Appendix A). Additionally, for each monomer, the active site is located at the interface of the subunit and the domain. The biological assembly of *h*OAT is a homodimer (Appendix A).

In the crystal structures of *h*OAT, inactivated by **4** (PDB code: 5VWO) and **8** (PDB code: 6V8D) [14,18], both compounds are covalently bonded to PLP, while their carboxylates interact with Tyr55 or Arg180. The structures of the *h*OAT-**5** co-crystal (PDB code: 8EZ1), shown in Figure 5, suggest covalent modification of Lys292 by the ligand in one of the three protein copies as a minor form (Figure 5, left) and a noncovalent form of the ligand in the two other protein copies (Figure 5, right) as a major form. The conjugated structure of the noncovalent form explains the maximum absorbance at 460 nm. It should also be noted that the carboxylate group forms salt bridges with Arg413 and hydrogen bonds with Gln266 rather than interacting with Arg180 or Tyr55. Arg413 is known to form a salt bridge with Glu235 in the native enzyme [27], and this interaction was found to be preserved in most of the *h*OAT inactivators, such as the complexes with **4** and **8**. In the case of noncovalent inhibition by **5**, the ligand is located 2.8 Å away from Lys292, resulting in the additional translational freedom for the carboxylate group to anchor with Arg413. The unique binding orientation of **5** also led to a water-mediated hydrogen bond interaction between the imine group of the ligand and Tyr55 residue.

Compound **5** has a unique combination of structural and mechanistic features that differentiate it from previous inactivators. These unique features result in the disruption of the Arg413-Glu235 salt bridge, which has been observed with *h*OAT inactivated by another MBI, **22** (Figure 6) [28]. However, structurally, **22** is longer and bulkier compared to known *h*OAT ligands. **22** occupies the the majority of the active site and forms hydrogen bonds with Tyr55, Arg413, and Gln266. In contrast, **5** is one of the smallest known *h*OAT inactivators, but its distinguishable inactivation mechanism allows it to orient itself in the active site in a way that it can also disrupt the Arg413-Glu235 salt bridge.

### 2.4. Transient-State Kinetics of hOAT Inactivation by 5

Due to the uniqueness of compound **5**, transient-state spectrophotometric measurements were performed to observe the kinetics of the inhibition of *h*OAT by **5** (Figure 6). Concentration dependences of the observed rate constants for individual phases were evaluated at 350, 420, and 580 nm. These wavelengths report principally on aldimine and quinonoid species. The data extracted at each wavelength were fit to Equation (4) in the experimental section that describes three exponential phases. At 350 nm, the observed rate constant for the first phase titrated hyperbolically to a limit. This indicates a rapid prior equilibrium for the association of **5** to form the first external aldimine species. The limiting rate constant for the forward decay of the external aldimine was 1.57 ± 0.15 s^−1^ and the fit to Equation (5) that describes a hyperbolic dependence with a positive y-intercept that defines the rate constant for the reverse reaction. This fit gave the dissociation constant (*k*_−1_/*k*_1_*)* for **5** to the *h*OAT internal aldimine of 1.44 ± 0.60 mM and a value for the rate constant for the decay of the external aldimine to the internal aldimine (*k*_−2_) of 0.17 ± 0.09 s^−1^ (Figure 7A).

The data extracted for 420 nm were fit by fixing the first phase to the *k_obs_* values determined for the first phase at 350 nm, where the first phase is well-delineated from the second. The second phase observed represents the decay of a second external aldimine-like species (see below) into a quinonoid with maximal absorption at 580 nm. The concentration dependence of the observed rate constant for the second phase gave a hyperbolic shape, indicating the reversibility of the prior step (*k*_2_ and *k*_−2_ in Figure 6). These data also fit best to Equation (5), and the limit of this dependence revealed a limiting forward rate constant (*k*_3_) of 0.059 ± 0.002 s^−1^ for the formation of the quinonoid intermediate. For the reverse reaction (*k*_−3_), the rate constant was 0.005 ± 0.002 s^−1^, Figure 7B). The data extracted for 580 nm report the formation and decay of the quinonoid and the decay of the subsequent diamine-like species (Figure 7). These data were fit by fixing the rate constant value for the final phase that represents the apparently irreversible decay of the diamine-like species to 0.002 s^−1^ (*k*_5_). This was done in part because the endpoint of the final phase was not definitively captured at the limit of the data collection (1250 s), and so no measure of error was obtained for the rate constant of this step. The fit reveals that the dependence is again hyperbolic due to the reversibility of the preceding step (*k*_3_, *k*_−3_), that it has a net rate constant (*k*_4_) of 0.046 ± 0.003 s^−1^ with no evidence for reversibility and was fit to Equation (6). Figure 6 depicts the summation of these observations.

Spliced charge-coupled device (CCD) datasets were fit using singular value decomposition (SVD) to deconvolute the data, yielding spectra for all species observed. The dataset obtained for 1960 µM of **5** was used to avoid complexity associated with enzyme precipitation at higher concentrations. The dataset fits best to an irreversible four-step model, consistent with data for a single-compound concentration that by themselves contain no evidence of reversibility. The rate constants obtained from the fit were as follows: *k*_2_ = 1.22 ± 0.02, *k*_3_ = 0.11 ± 0.01, *k*_4_ = 0.014 ± 0.01, *k*_5_ = 0.0023 ± 0.0002 (numbered according to Figure 6) (Figure 7). These rate constants are consistent with the values determined from analytical fits to exponentials at this concentration of inhibitor. In Figure 8, the time-zero spectrum for the *h*OAT internal aldimine was included for reference and but was not observed in the data. The time-zero spectrum was obtained by mixing *h*OAT with buffer (black dashed spectrum). The spectrum of the first species observed is consistent with that of an external aldimine (red spectrum). This species decays into a broader aldimine-like spectrum (green spectrum) before decaying to form a definitive quinonoid species (blue spectrum). The quinonoid decays to form a species largely devoid of visible wavelength absorption that is presumably a less conjugated diamine species (yellow spectrum). This species then decays to produce a third aldimine-like spectrum (cyan spectrum). 

### 2.5. The Inactivation Pathway of Compound 5 in hOAT

It should be noted that, in general, a quinonoid species (~580 nm) is thought to be formed right after the first external aldimine (~420 nm) (Figure 4, left). However, the transient-state measurement suggests that there is an additional broad aldimine-like species (green spectrum, Figure 8) observed between the quinonoid and the first external aldimine. We thus speculated that **External Aldimine I** (shown in Figure 7) may generate an enolate structure as **External Aldimine II**, which establishes dual interactions with Arg180 based on the molecular docking studies shown in Figure 9A. Free energy calculations in Figure 9B also support the existence of **External Aldimine II** tautomers, showing comparable results with **External Aldimine I**. Subsequently, **External Aldimine II** can undergo further tautomerization, triggered by the lone pair of electrons on Arg180, to form **Quinonoid I**. The elimination of the fluoride ion gives **External Aldimine III**, which can undergo attack by Lys292. This transient state further goes through the states of ***gem*-Diamines I and II** and releases enamine intermediate **Met I**. By mass spectrometry analysis, ketone intermediate **Met II** was detected as the predominant metabolite generated after incubating **5** with *h*OAT, validating the formation of **Met I** (Appendix A). The nucleophilic enamine intermediate **Met I** efficiently attacks the PLP–Lys292 **Internal Aldimine** and produces **Final Product I**, thereby inactivating the enzyme, which is covalently bound to PLP. Hydrolysis of the imine of **Final Product I** gives **Final Product II**. Elimination of Lys292 leads to **Final Product III** as the major form, which is in equilibrium with **Final Product I** (the minor form).

## 3. Materials and Methods

### 3.1. Chemistry

General Procedure. Commercially available reagents and solvents were used without further purification. All reactions were monitored by thin-layer chromatography (TLC) using 0.25 mm SiliCycle extra-hard 250 µM TLC plates (60 F254), and spots were visualized under UV (254 nm) and ceric ammonium molybdate or ninhydrin staining. Flash chromatography was performed on a Combi-Flash Rf system (Teledyne ISCO) with silica columns and reverse-phase C-18 columns. Analytical HPLC was used to determine the purity of all the final products using an Agilent 1260 series instrument with the following conditions: column, Phenomenex Kintex C-18 column (50 × 2.1 mm, 2.6 µm); mobile phase, 5–100% acetonitrile/water containing 0.05% TFA at a flow rate of 0.9 mL/min for 6 min; UV detection at 254 nm. ^1^H and ^13^C NMR spectra were obtained using the Bruker AVANCE III 500 MHz system and Bruker NEO console w/QCI-F cryoprobe 600 MHz system. Chemical shifts were reported relative to CDCl_3_ (δ = 7.26 for ^1^H NMR and δ = 77.16 for ^13^C NMR spectra) and CD_3_OD (δ = 3.31 for ^1^H NMR and δ = 49.15 for ^13^C NMR spectra). The following abbreviations for multiplicities were used: s = singlet; d = doublet; t = triplet; q = quartet; m = multiplet; dd = doublet of doublets; dt = doublet of triplets; dq = doublet of quartets, ddt = doublet of doublet of triplets, ddd = doublet of doublet of doublets; dddd = doublet of doublet of doublet of doublets; dddt = doublet of doublet of doublet of triplets; br s = broad singlet. Low-resolution mass spectra (LRMS) were obtained using a Thermo TSQ Quantum system in the positive ion mode using atmospheric pressure chemical ionization (APCI)/electrospray ionization (ESI) with a reverse-phase Agilent Infinity 1260 HPLC system. High-resolution mass spectra (HRMS) were obtained on an Agilent 6210 LC-TOF spectrometer in the positive ion mode using electrospray ionization (ESI) with an Agilent G1312A HPLC pump and an Agilent G1367B autoinjector at the Integrated Molecular Structure Education and Research Center (IMSERC), Northwestern University. The purity of all tested compounds for in vitro biological studies was >95%, assessed by HPLC analysis.

**(1*R*,4*R*,6*S*,7*R*)-7-Bromo-6-hydroxy-2-(4-methoxybenzyl)-2-azabicyclo[2.2.1]hep-tan-3-one (12)** was synthesized from (1*R*)-(−)-2-azabicyclo[2.2.1]hept-5-en-3-one (**(-)-10**, CAS#: 79200-56-9, 30 g, 270 mmol) following the procedure published previously [29]. The desired product **12** was afforded as an off-while solid (27.3 g, 31% over two steps). ^1^H NMR (500 MHz, CDCl_3_) δ 7.19–7.12 (m, 2H), 6.92–6.85 (m, 2H), 4.58 (d, *J* = 14.7 Hz, 1H), 4.24 (t, *J* = 1.7 Hz, 1H), 3.98 (d, *J* = 14.7 Hz, 1H), 3.96–3.92 (m, 1H), 3.71 (s, 1H), 2.95 (dd, *J* = 3.8, 1.9 Hz, 1H), 2.49–2.37 (m, 2H), 2.15 (dt, J = 13.8, 3.8 Hz, 1H). ^13^C NMR (126 MHz, CDCl_3_) δ 172.6, 159.6, 129.7 (2C), 128.0, 114.6 (2C), 65.7, 55.5, 50.8, 50.5, 44.1, 33.7. LRMS (APCI) calc. for C_14_H_17_BrNO_3_ [M+H]^+^: 326.04; found, 326.21; T_R_ = 2.00 min.

**(1*R*,4*R*,6*S*,7*R*)-7-Bromo-6-fluoro-2-(4-methoxybenzyl)-2-azabicyclo[2.2.1]heptan-3-one (13)**. To a stirred solution of **12** (1000 mg, 3.07 mmol) in DCM (20 mL) was added DAST (0.41 mL, 4.60 mmol) at 0 °C. The resulting mixture was slowly warmed to room temperature and stirred overnight. After the completion of the reaction, the solution was quenched with water and extracted with DCM (30 mL × 3). The combined organic layers were separated, washed with brine, dried over Na_2_SO_4_, and concentrated under vacuum conditions. The crude product was purified via Combi-Flash chromatography (EtOAc/hexane: 0–100%) to yield **13** as a colorless oil (900 mg, 90%). ^1^H NMR (500 MHz, CDCl_3_) δ 7.16 (d, *J* = 8.3 Hz, 2H), 6.89 (d, *J* = 8.4 Hz, 2H), 4.64 (dddd, *J* = 53.8, 7.3, 2.8, 1.4 Hz, 1H), 4.49 (d, *J* = 14.7 Hz, 1H), 4.21 (s, 1H), 4.07 (d, *J* = 14.7 Hz, 1H), 3.90 (s, 1H), 3.82 (s, 3H), 2.97–2.88 (m, 1H), 2.45 (ddt, *J* = 27.8, 13.9, 3.5 Hz, 1H), 2.36–2.25 (m, 1H). ^13^C NMR (126 MHz, CDCl_3_) δ 172.9, 159.7, 129.8 (2C), 127.8, 114.7 (2C), 90.77 (d, *J* = 200.1 Hz), 64.50 (d, *J* = 22.8 Hz), 55.5, 50.52 (d, *J* = 2.4 Hz), 48.05 (d, *J* = 1.9 Hz), 44.4, 31.10 (d, *J* = 21.6 Hz). LRMS (APCI) calc. for C_14_H_16_BrFNO_2_ [M+H]^+^: 328.03; found, 328.10; T_R_ = 2.62 min.

**(1*R*,4*R*,6*S*,7*R*)-*tert*-Butyl 7-Bromo-6-fluoro-3-oxo-2-azabicyclo[2.2.1]heptane-2-carboxylate (14)**. (i) To a stirred solution of **13** (900 mg, 2.75 mmol) in CH_3_CN (40 mL) was added an aqueous solution of ceric ammonium nitrate (4.52 g in 12 mL water, 8.26 mmol) at room temperature. The resulting mixture was stirred for 2 h until the starting material disappeared. After the completion of the reaction, the reaction mixture was concentrated under vacuum conditions, and the residue was extracted with EtOAc (25 mL × 3). The combined organic layers were separated, washed with sat. Na_2_CO_3_ solution and brine, dried over Na_2_SO_4_, and concentrated under vacuum conditions. The crude product was purified via Combi-Flash chromatography (EtOAc-hexane: 0–100%) to lactam intermediate (330 mg, 1.60 mmol) as an off-white solid. (ii) To a stirred solution of lactam intermediate (330 mg, 1.60 mmol) in DCM (20 mL) were added DIPEA (0.33 mL, 1.91 mmol), DMAP (20 mg, 0.16 mmol), and Boc_2_O (381 mg, 1.75 mmol) at room temperature, and the resulting mixture was stirred at room temperature for 3 h. After the completion of the reaction, the solution was quenched with sat. NH_4_Cl solution and extracted with DCM (25 mL × 3). The combined organic layers were separated, washed with brine, dried over Na_2_SO_4_, and concentrated under vacuum conditions. The crude product was purified via Combi-Flash chromatography (EtOAc/Hexane: 0–60%) to yield **14** (480 mg, 56% over two steps) as a white solid. ^1^H NMR (500 MHz, CDCl_3_) δ 4.95 (dddd, *J* = 52.9, 5.9, 3.1, 1.6 Hz, 1H), 4.75 (s, 1H), 4.32 (t, *J* = 1.6 Hz, 1H), 3.00 (dt, *J* = 4.0, 1.7 Hz, 1H), 2.54 (ddt, *J* = 28.3, 14.3, 3.6 Hz, 1H), 2.47–2.39 (m, 1H), 1.53 (s, 9H). ^13^C NMR (126 MHz, CDCl_3_) δ 169.9, 147.8, 90.77 (d, *J* = 199.6 Hz), 84.8, 64.12 (d, *J* = 24.3 Hz), 51.82 (d, *J* = 2.5 Hz), 45.50 (d, *J* = 1.9 Hz), 30.84 (d, *J* = 22.0 Hz), 28.1 (3C). LRMS (APCI) calc. for C_11_H_16_BrFNO_3_ [M+H]^+^: 308.03; found, 308.26; T_R_ = 2.70 min.

**(3*S*,4*S*)-Methyl 3-((*tert*-Butoxycarbonyl)amino)-4-fluorocyclopent-1-enecarbo-xylate (15)**. To a stirred solution of **14** (480 mg, 1.56 mmol) in MeOH (20 mL) was added K_2_CO_3_ (647 mg, 4.69 mmol) at room temperature. Then, the resulting mixture was stirred for an additional 1 h and then neutralized with sat. NH_4_Cl solution and extracted with EtOAc (25 mL × 3). The combined organic layers were separated, washed with brine, dried over Na_2_SO_4_, and concentrated under vacuum conditions. The crude product was purified via Combi-Flash chromatography (EtOAc/Hexane: 0–60%) to yield **15** (190 mg, 47%) as a white solid. ^1^H NMR (500 MHz, CDCl_3_) δ 6.58 (s, 1H), 5.20–4.98 (m, 1H), 4.80 (d, *J* = 21.3 Hz, 1H), 4.58 (s, 1H), 3.77 (s, 3H), 3.06 (dddt, *J* = 22.1, 18.0, 6.4, 1.9 Hz, 1H), 2.76 (dddt, *J* = 26.9, 17.9, 2.8, 1.3 Hz, 1H), 1.45 (s, 9H). ^13^C NMR (126 MHz, CDCl_3_) δ 164.5, 154.9, 138.8, 136.5, 97.60 (d, *J* = 184.1 Hz), 80.51, 62.86 (d, *J* = 33.0 Hz), 52.1, 37.43 (d, *J* = 24.8 Hz), 28.5 (3C). LRMS (APCI) calc. for C_7_H_10_FNO_2_ [M–Boc+H]^+^: 159.07; found, 158.91; T_R_ = 2.56 min.

**(3*S*,4*S*)-3-Amino-4-fluorocyclopent-1-enecarboxylic acid hydrochloride (5)**. To a stirred solution of **15** (190 mg, 0.73 mmol) in acetic acid (7 mL) was added 4N HCl (7 mL) at room temperature, and the resulting mixture was heated at 70 °C overnight. After the completion of the reaction, the excess solvent was removed under vacuum conditions. The crude product was washed with acetonitrile to afford **5** (130 mg, 98%) as an off-white solid. ^1^H NMR (500 MHz, CD_3_OD) δ 6.55 (s, 1H), 5.33 (ddt, *J* = 51.0, 6.4, 3.0 Hz, 1H), 4.54 (dtt, *J* = 22.1, 2.8, 1.5 Hz, 1H), 3.22 (dddt, *J* = 19.9, 18.4, 7.0, 2.0 Hz, 1H), 2.83 (dddt, *J* = 27.6, 18.3, 3.2, 1.5 Hz, 1H). ^13^C NMR (126 MHz, MeOD) δ 166.1, 142.1, 134.0, 95.99 (d, *J* = 183.6 Hz), 63.29 (d, *J* = 31.4 Hz), 38.75 (d, *J* = 24.5 Hz). HRMS (ESI) calc. for C_6_H_9_FNO_2_ [M+H]^+^: 146.0612; found, 146.0613.

**(±)-*tert*-Butyl 3-Oxo-2-azabicyclo[2.2.1]hept-5-ene-2-carboxylate (16)**. To a stirred solution of (±)-2-azabicyclo[2.2.1]hept-5-en-3-one (**(±)-10**, CAS# 49805-30-3, 2.5 g, 23 mmol) in DCM (250 mL) were added Boc_2_O (6.0 g, 27.5 mmol), TEA (3.83 mL, 27.5 mmol), and DMAP (280 mg, 2.3 mmol). The resulting mixture was stirred at room temperature overnight. After the completion of the reaction, the solution was quenched with water and extracted with DCM (25 mL × 3). The combined organic layers were separated, washed with brine, dried over Na_2_SO_4_, and concentrated under vacuum conditions. The crude product was purified via Combi-Flash chromatography (EtOAc/hexane: 0–100%) to give **16** (4.5 g, 94%) as an off-white solid. ^1^H NMR (500 MHz, CDCl_3_) δ 6.87 (dd, *J* = 5.3, 2.4 Hz, 1H), 6.64 (dq, *J* = 5.3, 1.6 Hz, 1H), 4.94 (s, 1H), 3.36 (s, 1H), 2.39–2.26 (m, 1H), 2.13 (dt, *J* = 8.5, 1.3 Hz, 1H), 1.48 (s, 9H). ^13^C NMR (126 MHz, CDCl_3_) δ 176.3, 150.5, 140.1, 138.3, 82.7, 62.5, 55.0, 54.6, 28.2 (3C). LRMS (APCI) calc. for C_11_H_16_NO_3_ [M+H]^+^: 210.11; found, 209.93.

**(±)-*tert*-Butyl 7-Oxo-3-oxa-6-azatricyclo[3.2.1.02,4]octane-6-carboxylate (17)**. To a stirred solution of **16** (4.5 g, 21.5 mmol) in CHCl_3_ (150 mL) was added *m*-CPBA (7.2 g, ~77%, 32.3 mmol) at room temperature under an argon atmosphere. The resulting mixture was stirred at 65 °C overnight. After the completion of the reaction, the solution was quenched with sat. NaS_2_O_3_ solution and extracted with CHCl_3_ (100 mL × 3). The combined organic layers were separated, washed with sat. Na_2_CO_3_ solution and brine, dried over Na_2_SO_4_, and concentrated under vacuum conditions. The crude product was purified via Combi-Flash chromatography (EtOAc/hexane: 0–100%) to yield **17** (2.3 g, 48%) as an off-white solid. ^1^H NMR (500 MHz, CDCl_3_) δ 4.61 (p, *J* = 1.6 Hz, 1H), 3.77 (dd, *J* = 3.6, 1.2 Hz, 1H), 3.61 (dd, *J* = 3.7, 1.5 Hz, 1H), 3.06 (p, *J* = 1.6 Hz, 1H), 1.81 (dt, *J* = 10.4, 1.7 Hz, 1H), 1.70–1.61 (m, 1H), 1.52 (s, 9H). ^13^C NMR (126 MHz, CDCl_3_) δ 173.5, 149.9, 83.5, 59.1, 53.3, 50.1, 48.5, 28.2 (3C), 27.2. LRMS (APCI) calc. for C_11_H_16_NO_4_ [M+H]^+^: 226.11; found, 226.04.

**Methyl *trans*-4-((*tert*-Butoxycarbonyl)amino)-3-hydroxycyclopent-1-enecarboxylate (18).** To a stirred solution of **17** (1.0 g, 4.44 mmol) in MeOH (30 mL) was added K_2_CO_3_ (1.84 g, 13.3 mmol). The resulting mixture was stirred at room temperature for 1 h. After the completion of the reaction, the solution was quenched with sat. NH_4_Cl solution and extracted with DCM (25 mL × 3). The combined organic layers were separated, washed with sat. Na_2_CO_3_ solution and brine, dried over Na_2_SO_4_, and concentrated under vacuum conditions. The crude product was purified via Combi-Flash chromatography (EtOAc/hexane: 0–100%) to yield **18** (0.9 g, 79%) as a white solid. ^1^H NMR (500 MHz, DMSO-*d*_6_) δ 7.14 (d, *J* = 7.8 Hz, 1H), 6.50 (q, *J* = 1.9 Hz, 1H), 5.31 (d, *J* = 6.6 Hz, 1H), 4.59 (tq, *J* = 6.0, 2.0 Hz, 1H), 3.78 (p, *J* = 8.2, 7.2 Hz, 1H), 3.67 (s, 3H), 2.76 (dd, *J* = 16.1, 8.2 Hz, 1H), 2.21 (ddt, *J* = 16.2, 6.7, 2.1 Hz, 1H), 1.39 (s, 9H). ^13^C NMR (126 MHz, DMSO-*d*_6_) δ 164.5, 155.3, 143.8, 133.6, 79.7, 77.7, 59.3, 51.5, 36.1, 28.2 (3C). LRMS (APCI) calc. for C_7_H_11_NO_3_ [M–Boc+H]^+^: 157.07; found, 157.59.

**Methyl *trans*-4-(Bis(4-methoxybenzyl)amino)-3-hydroxocyclopent-1-ene-1-carboxylate (19)**. (i) To a stirred solution **18** (3.00 g, 11.7 mmol) in MeOH (15 mL) was added 3M HCl in MeOH (30 mL). The resulting mixture was stirred at room temperature under an argon atmosphere for 1 h. After the completion of the reaction, the excessive solution was removed under vacuum conditions. The crude product was used directly in the next step. (ii) To a stirred solution of the crude product (theoretically 2.257 g, 11.66 mmol) in DCE (72.9 mL) were added TEA (3.25 mL, 23.332 mmol) and 4-anisaldehyde (4.26 mL, 34.98 mmol). The resulting mixture was stirred under an argon atmosphere at room temperature until all the starting material was dissolved in solution, followed by the addition of AcOH (2.67 mL, 46.65 mmol). The resulting mixture was then stirred at 75 °C for an additional 1 h under an argon atmosphere. After that, the reaction solution was then cooled to room temperature, and NaBH(OAc)_3_ (7.412 g, 34.98 mmol, 3.0 equiv) was slowly added to the solution. The resulting mixture was then stirred at 75 °C under an argon atmosphere overnight. After the completion of the reaction, the solution was quenched with sat. NH_4_Cl solution and extracted with DCM (25 mL × 3). The combined organic layers were separated, washed with sat. Na_2_CO_3_ solution and brine, dried over Na_2_SO_4_, and concentrated under vacuum conditions. The crude product was purified via Combi-Flash chromatography (EtOAc/hexane: 0–100%) to yield **19** (2.7 g, 58% over two steps) as a colorless oil. ^1^H NMR (500 MHz, CDCl_3_) δ 7.31–7.26 (m, 4H), 6.87–6.81 (m, 4H), 6.60 (q, *J* = 1.8 Hz, 1H), 4.99–4.89 (m, 1H), 3.79 (s, 6H), 3.74 (s, 3H), 3.68 (d, *J* = 13.8 Hz, 2H), 3.49 (d, *J* = 13.7 Hz, 2H), 3.41 (q, *J* = 7.6 Hz, 1H), 2.73 (ddt, *J* = 16.8, 8.4, 1.8 Hz, 1H), 2.63–2.50 (m, 1H). ^13^C NMR (126 MHz, CDCl_3_) δ 165.2, 159.4 (2C), 142.4, 133.3, 130.1 (2C), 128.8 (4C), 114.1 (4C), 69.6, 65.2, 55.5 (2C), 55.4 (2C), 52.0, 30.6. LRMS (APCI) calc. for C_23_H_28_NO_5_ [M+H]^+^: 398.20; found, 398.02.

**Methyl *trans*-4-(Bis(4-methoxybenzyl)amino)-3-fluorocyclopent-1-enecarboxylate (20)**. To a stirred solution of **19** (1.00 g, 2.52 mmol) in DCM (30 mL) was added Deoxo-Fluor (1.40 mL, 3.77 mmol) dissolved in DCM (20 mL) at –78 °C under an argon atmosphere. The resulting mixture was warmed to room temperature and stirred overnight. After the completion of the reaction, the solution was quenched with water and extracted with DCM (25 mL × 3). The combined organic layers were separated, washed with sat. Na_2_CO_3_ solution and brine, dried over Na_2_SO_4_, and concentrated under vacuum conditions. The crude product was purified via Combi-Flash chromatography (EtOAc/hexane: 0–100%) to yield **20** (0.26 g, 26%) as a colorless oil. ^1^H NMR (500 MHz, CDCl_3_) δ 7.35–7.18 (m, 4H), 6.92–6.78 (m, 4H), 6.64 (s, 1H), 5.77 (dd, *J* = 53.6, 2.4 Hz, 1H), 3.80 (s, 6H), 3.76 (s, 3H), 3.72–3.65 (m, 1H), 3.56 (s, 4H), 2.88–2.76 (m, 1H), 2.63–2.51 (m, 1H). ^13^C NMR (126 MHz, CDCl_3_) δ 164.8, 158.9 (2C), 139.75 (d, *J* = 9.4 Hz), 137.72 (d, *J* = 19.4 Hz), 131.1 (2C), 129.9 (4C), 113.8 (4C), 98.79 (d, *J* = 177.7 Hz), 66.18 (d, *J* = 20.8 Hz), 55.4 (2C), 54.4 (2C), 52.1, 32.52 (d, *J* = 4.1 Hz). LRMS (APCI) calc. for C_23_H_27_FNO_5_ [M+H]^+^: 400.19; found, 400.14.

**Methyl *trans*-4-((*tert*-Butoxycarbonyl)amino)-3-fluorocyclopent-1-enecarboxylate (21)** (i) To a stirred solution of **20** (0.33 g, 0.829 mmol) in CH_3_CN (15 mL) was added ceric ammonium nitrate (3.634 g, 6.63 mmol) dissolved in H_2_O (5 mL) at 0 °C. The resulting mixture was stirred at room temperature for 3 h. After the completion of the reaction, the solution was diluted with EtOAc (50 mL) and NaHCO_3_ (sat., aq.) and stirred vigorously until the pH of the aqueous layer was 8. The suspension was filtered through a Celite pad and washed with EtOAc. The filtrate was then concentrated under vacuum conditions to afford a solution (presumed aqueous from residual water) containing deprotected product. This crude product was used directly in the next step without further purification. (ii) To a stirred solution of the crude product (theoretically 0.132 g, 0.829 mmol) in MeOH (9 mL) was added Boc_2_O (0.271 g, 1.24 mmol). The resulting mixture was stirred at room temperature overnight. After the completion of the reaction, the remaining solvent was concentrated under vacuum conditions and via Combi-Flash chromatography (EtOAc/hexane: 0–100%) to afford **21** as a colorless oil (60 mg, 28%). ^1^H NMR (500 MHz, CDCl_3_) δ 6.65 (t, *J* = 2.2 Hz, 1H), 5.61 (d, *J* = 53.4 Hz, 1H), 4.80 (br s, 1H), 4.29–4.14 (m, 1H), 3.77 (s, 3H), 3.18–3.04 (m, 1H), 2.51 (s, 1H), 1.45 (s, 9H). ^13^C NMR (126 MHz, CDCl_3_) δ 164.5, 155.3, 139.76 (d, *J* = 9.6 Hz), 136.86 (d, *J* = 18.7 Hz), 101.25 (d, *J* = 180.6 Hz), 80.3, 57.35 (d, *J* = 24.9 Hz), 52.2, 36.7, 28.5 (3C). LRMS (APCI) calc. for C_7_H_10_FNO_2_ [M–Boc+H]^+^: 159.07; found, 159.99.

***trans*-4-Amino-3-fluorocyclopent-1-enecarboxylic acid hydrochloride (6)**. To a stirred solution of **21** (30 mg, 0.72 mmol) in acetic acid (7 mL) was added 4N HCl (7 mL) at room temperature. The resulting mixture was heated at 70 °C overnight. After the completion of the reaction, the excess solvent was removed under vacuum conditions. The crude product was purified via Combi-Flash chromatography (C18 reverse-phase column, CH_3_CN/H_2_O: 0–5%) to give **6** as a white powder HCl salt (5 mg, 24%). ^1^H NMR (500 MHz, D_2_O) δ 6.69 (dq, *J* = 4.1, 2.2 Hz, 1H), 5.90 (dddt, *J* = 52.3, 3.2, 2.3, 1.0 Hz, 1H), 4.14 (dddd, *J* = 23.8, 8.5, 5.5, 4.2 Hz, 1H), 3.33–3.17 (m, 1H), 2.73–2.55 (m, 1H). ^13^C NMR (126 MHz, D_2_O) δ 167.8, 140.51 (d, *J* = 9.6 Hz), 135.12 (d, *J* = 19.7 Hz), 99.25 (d, *J* = 178.3 Hz), 55.40 (d, *J* = 26.1 Hz), 34.4. HRMS (ESI) calc. for C_6_H_9_FNO_2_ [M+H]^+^: 146.0612; found, 146.0613.

### 3.2. Expression and Purification of hOAT 

*h*OAT was expressed and purified using previously published protocols [18]. Briefly, *E. coli* BL21(DE3) cells, containing the pMAL-t-*h*OAT plasmid, were incubated at 37 °C with shaking in a lysogeny broth (LB) medium supplemented with 100 µg/mL ampicillin. When the culture OD_600_ reached a value of 0.7, expression of the MBP−t-*h*OAT fusion protein was induced by the addition of 0.3 mM isopropyl β-D-1-thiogalactopyranoside and further incubated for an additional 16−18 h at 25 °C. Cells were harvested by centrifugation, washed with buffer A comprised of 20 mM Tris-HCl, 200 mM NaCl, and 100 µM PLP, pH 7.4, and flash-frozen in liquid nitrogen and stored at −80 °C. The frozen cell pellet was then thawed, sonicated in buffer A, and centrifuged at 40,000× *g* for 20 min. The resulting supernatant was loaded onto an amylose affinity column pre-equilibrated with buffer A. The column was washed thoroughly, and the MBP−t-*h*OAT fusion protein was eluted from the column with 10 mM maltose. Fractions containing the fusion protein were combined and treated with TEV protease to remove the MBP tag. The cleaved *h*OAT protein was collected and concentrated using a centrifugal filter. The protein was then further purified by size exclusion chromatography using a HiLoad Superdex-200PG column. The column was pre-equilibrated, and the protein eluted in buffer containing 50 mM HEPES, 100 µM PLP, and 300 mM NaCl, pH 7.5. 

### 3.3. Aminotransferases and Coenzymes for Kinetic Studies

All reagents for enzyme purification and assays were purchased from Sigma-Aldrich (St. Louis, MO, USA). Human OAT (0.672 mg/mL) was purified from *E. coli* BL21 (DE3) cells following the procedure described above, and coenzyme human recombinant pyrroline 5-carboxylate reductase (PYCR1) was expressed, grown, and purified according to a procedure outlined in the literature [30]. γ-Aminobutyric acid aminotransferase (GABA-AT, 0.4 mg/mL) was purified from pig brain following a procedure described previously [31], and coenzyme succinic semialdehyde dehydrogenase (SSDH) was purified from GABase (catalog No. G7509-25UN, Sigma-Aldrich), a commercially available mixture of SSDH and GABA-AT, using a known procedure [32]. Coupled enzyme assays for GABA-AT and *h*OAT were carried out according to previous procedures [33,34]. All of the enzyme assays for the inactivation, partition ratio, and dialysis experiment were recorded on a Synergy H1 hybrid multimode microplate reader (Biotek, Winooski, VT, USA) with transparent 96-well plates.

### 3.4. Denaturing Intact Protein and Small Molecule Mass Spectrometry

Treated and unmodified purified *h*OAT samples were desalted ten times with Optima-grade water (Fisher, Loughborough, UK) on Amicon Ultra 10 kDa molecular weight spin filters (Millipore, Darmstadt, Germany). To chromatographically resolve protein, 0.5 µg of protein was loaded onto a 3 cm PLRP-S (Agilent, Santa Clara, CA, USA) trap column using a Dionex Ultimate3000 liquid chromatography system (Thermo Fisher, Waltham, MA, USA). The protein analyte was washed with a 10 min isocratic gradient of 10% Solvent B (95% acetonitrile/5% H_2_O/0.2% formic acid) and 90% Solvent A (5% acetonitrile/95% H_2_O/0.2% formic acid). Protein was resolved on an in-house-made 75 µm ID × 15 cm long nanopore capillary column packed with PLRP-S resin (Agilent). The LC system was operated at a flow rate of 0.3 µL/min at the following gradient: 0–10 min 10% Solvent B; 10–12 min to 40% Solvent B; 12–22 min to 90% Solvent B; 22–24 min at 90% Solvent B; 24–26 min to 10% Solvent B; 26–30 min isocratic at 10% Solvent B. *h*OAT samples were introduced into a Thermo Fisher Orbitrap Fusion Lumos or Eclipse mass spectrometer, and full MS data were acquired as previously described [28]. Small molecule *h*OAT substrate and product masses were identified and characterized by positive and negative mode high-resolution LC-MS/MS on a Q-Exactive Orbitrap mass spectrometer (Thermo), as previously described [13,14,28].

### 3.5. Theoretical pK_a_ Calculations

Theoretical p*K_a_* calculations were conducted following a procedure described previously [14,26]. The geometries of the neutral and deprotonated species of **M15**, **M15-1**, and **M15-2** were fully optimized using the DFT B3LYP/6-31G** level of theory. For all of the investigated compounds, the gas-phase Gibbs free energy changes ($\Delta G_{g}^{o}$) of compounds were calculated using Gaussian09 software [35]. The solvation free energies were calculated by applying polarizable continuum model (PCM), using the same level of theory and basis set (B3LYP/6-31G**) which was used for geometry determination in the gas phase. The PCM calculations were used with the UAHF atomic radii when building the solvent cavity to calculate the Gibb’s free energy of solvation. The *p*K_a_ values were obtained applying the following Equations (1)–(3) and the thermodynamic cycle A, presented by Ghalami-Choobar and coworkers [25].
(1)$$\Delta G_{\mathrm{aq}}^{o}=\Delta G_{g}^{o}+\Delta G_{s}^{o}$$
(2)$$\Delta G_{\mathrm{aq}}^{o}=-2.303\mathrm{RT}\log K_{a}$$
(3)$$\Delta G_{\mathrm{aq}}^{o}=\Delta G_{g}^{o}\left(A^{-}\right)+\Delta G_{s}^{o}\left(H^{+}\right)-\Delta G_{s}^{o}\left(\mathrm{AH}\right)+G_{g}^{o}\left(A^{-}\right)+G_{s}^{o}\left(H^{+}\right)-G_{s}^{o}\left(\mathrm{AH}\right)$$

### 3.6. Electrostatic Potential (ESP) Charge Calculation

The three-dimensional (3D) molecular models of **M18**, **M18-1**, and **M18-2** were built up using Spartan’14 software (Wavefuction, Inc., 2014, Irvine, United States). The build structures were refined by molecular mechanics using Merck molecular force field (MMFF94). Then, with the lowest energy conformer selected, the equilibrium geometry and molecular orbitals were calculated using Hartree–Fock (HF) at the 6-31G* level of theory. Spartan’14 was also used to generate electron density and electrostatic potential maps.

### 3.7. Co-Crystallization of hOAT with Compound **5**


Crystal Growth. The freshly prepared enzyme was dialyzed with 50 mM potassium pyrophosphate which contained 5 mM α-ketoglutarate at pH 8.0 overnight. The enzyme was then incubated with excess amount of compound **5** at 4 °C overnight. The pre-inactivated *h*OAT sample was buffer exchanged into 50 mM Tricine pH 7.8 and concentrated to a protein concentration of 6 mg/mL. Previously reported crystallization conditions [28] were optimized using the hanging drop vapor diffusion method by varying PEG 6000 (8–12%), NaCl (100–250 mM), glycerol (0–10%) with 100 mM Tricine pH 7.8 as the buffer. For each hanging drop, either 2 or 3 µl of protein solution was mixed with equal volume of well solution. Several rounds of seeding followed to improve crystal size and quality. The crystals with the best morphology and size grew in a final condition containing 10% PEG 6000, 100 mM NaCl, 10% glycerol, 100 mM Tricine pH 7.8 at the temperature of 20 °C (293 K). Crystals were transferred to a cryo-protectant solution (well solution supplemented with 30% glycerol) before being flash-frozen in liquid nitrogen. 

X-ray Diffraction and Data Processing. Monochromatic X-ray diffraction data were collected at the LS-CAT beamline 21-ID-D at the Advanced Photon Source at Argonne National Laboratory. Data were collected at a wavelength of 1.127 Å and a temperature of 100 K using a Dectris Eiger 9M detector. Datasets were processed and analyzed with autoPROC software [36]. 

Model Building and Refinement. The *h*OAT structure was solved by molecular replacement using PHASER [37] in Phenix. The starting search model was the previously published structure of *h*OAT (PDB code: 1OAT [27]). The model building and refinement were accomplished in Coot [38] and Phenix [39], respectively, as an iterative process until the lowest possible R_free_/R factor values were attained. Structural depiction figures were prepared using UCSF Chimera [40].

### 3.8. Transient-State Methods

The reaction of **5** with OAT was observed in a transient state using stopped-flow spectrophotometry (TgK Scientific, Wiltshire, UK). Solutions were mixed in a 1:1 ratio, and spectrophotometric data were collected using a charged coupled device for wavelengths (260–800 nm). To capture data with temporal resolution that adequately described both fast and slow chemical steps, duplicate datasets were collected using a log time base for two timeframes (0.0025–5 seconds and 0.0025–1250 seconds). The duplicate datasets for any one timeframe were averaged and the data for both time frames were spliced together at 5 seconds to form one dataset. In these experiments, 13.4 µM *h*OAT was allowed to react with varied concentrations of **5** (62, 123, 245, 490, 980, 1960, 3920 and 7830 µM) in 50 mM HEPES, 200 mM NaCl, pH 7.5 at 10 °C (all concentration indicated are post-mixing). Reaction traces at specific wavelengths were extracted from the spectral data and combined into single datasets that depict the concentration dependence of the processes observed. These data were fit analytically to a linear combination of exponentials (Equation (4)) to evaluate concentration dependences. In this equation, A_Xnm_ is the absorbance at any time, DA_n_ is the amplitude associated with phase n, *k_n_* is the rate constant for phase n, and C is the absorption at infinite time. The dependencies of observed rate constants were fit to Equation (5) for reversible steps and to Equation (6) for irreversible steps that follow reversible steps. In these equations *k_obs_* is the observed rate constant, *k_n_* is the limiting value of the forward rate constant, [5] is the concentration of inhibitor, and K_eq_ is the equilibrium constant for the preceding step. Hybrid timeframe CCD spectral datasets, collected at a concentration of **5** equal to 1960 µM, were fit globally to an irreversible linear four-step model using the Spectrafit singular value decomposition (SVD) module of the KinTek Explorer software. This concentration was selected to avoid the enzyme precipitation that was evident in the data collected for 3920 and 7830 µM of **5**.
(4)$$A_{Xnm}=\Delta A_{1}\left(e^{-k_{1}t}\right)+\Delta A_{2}\left(e^{-k_{2}t}\right)+\Delta A_{3}\left(e^{-k_{3}t}\right)+C$$
(5)$$k_{obs}=k_{-n}+\frac{k_{n}\left[5\right]}{(1/K_{eq}+\left[5\right]}$$
(6)$$k_{obs}=\frac{k_{n}\left[5\right]}{(1/K_{eq}+\left[\;5\right]}$$

### 3.9. Gibb’s Free Energy Calculation

MOPAC 2016 is a computational chemistry software that is based on the concepts of quantum theory and thermodynamics, using some concepts of advanced mathematics. It is a semi-empirical molecular orbital package used for the study of solid-state nanostructure molecular structures, and their reactions [41]. In this context, MOPAC 2016 has been used for setting the molecular geometries of each tautomeric form in Figure 9B, followed by optimization through the PM7 semi-empirical method. Solvation by water molecules was not considered for the calculation. A combination of molecular mechanics energy, polar and nonpolar energies, and entropy have been considered for the resulting binding free energy of each tautomer presented in Figure 9B. Enthalpy (ΔH°) and entropy (ΔS°) contribute to the ΔG° value of each tautomer, something also determined by the Gibb’s free energy equation: (7)$$\Delta G^{o}=\Delta H^{o}-T\Delta S^{o}$$

### 3.10. Molecular Docking Protocol

Molecular docking studies were conducted following a procedure described previously [13,26]. Docking models of ligands bound to *h*OAT were developed using the Molecular Operating Environment (MOE) computational suite’s Builder utility [42]. The energy minimization of ligands was conducted in the gas phase using the force field MMFF94X, followed by the Conformational Search protocol to generate structural–conformation databases. The X-ray crystal structures of native *h*OAT (1OAT) and inactivated *h*OAT (1GBN) were uploaded to MOE, followed by the use of the Receptor Preparation step. The tight-binding product in the active pocket was deleted, and catalytic Lys292 was neutralized. The docking site was specified by the Lys292-PLP linkage. Ligand docking simulation was carried out in the prepared aminotransferase enzyme models with unrelated substrates and the solvent atoms inactivated. Ligand placement employed the Alpha Triangle method with Affinity ΔG scoring, generating 300 data points that were further refined using the induced fit method with GBVI/WSA ΔG scoring to obtain the top 50 docking results. The docking results of each ligand were analyzed for selection of the best docking pose, based on the score and reported X-ray structures. All renderings were then performed in MOE. The PLP moieties of the docked molecules (Figure 9A) showed comparable docking poses with our previous findings [13,14], while carboxylate groups formed tight interactions with Arg180, indicating reasonable docking poses in the active site of *h*OAT.

## 4. Conclusions

The incorporation of a double bond into a cyclopentane system has significantly improved the inactivation efficiency of our recent MBIs by enhancing the reactivity of the key C_γ_ hydrogen and providing conformational changes [26,43,44,45]. In this work, we carried out an integrated mechanistic study with *h*OAT and **5**, demonstrating that the addition of a double bond also influences the inactivation mechanism pathways. Unlike the parent monofluorinated cyclopentane (**4**), showing a typical enamine addition mechanism, the α, β-unsaturated cyclopentene **5** mainly inactivates *h*OAT through noncovalent, tight-binding interactions. This has been fully characterized by intact protein MS, wavelength absorbance measurements of the final products, and co-crystallization. The co-crystal complex also reveals for the first time that, in both the minor covalent and major noncovalent forms, the carboxylate group of **5** establishes unusual salt bridges with Arg413 rather than salt bridges with Arg180/Tyr55, as was previously observed in all the co-crystals of *h*OAT with other MBIs [14,26,28,45]. The transition state kinetics further indicate that a secondary external aldimine is formed between the first external aldimine and the first quinonoid transitions. Supported by molecular docking and free energy calculations, we propose the potential existence of an enolate intermediate that might be converted from the carboxylate of **5** and participate in the inactivation cascade directly. Overall, compound **5** inactivates *h*OAT through a unique enamine addition mechanism. These results will aid in the future design of selective *h*OAT inactivators.

## Data Availability

Not applicable.

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
