# Peer review of "Structural and Mechanistic Basis for the Inactivation of Human Ornithine Aminotransferase by (3S,4S)-3-Amino-4-fluorocyclopentenecarboxylic Acid"

_molecules, 2023, doi:10.3390/molecules28031133_

Round 1

Reviewer 1 Report

Manuscript ID:

Title: “Structural and Mechanistic Basis for the Inactivation of Human 2 Ornithine Aminotransferase by (S)-3-Amino-(S)-4-fluorocyclo- 3 pentenecarboxylic Acid” by Richard B. Silverman et al.

The authors have carried out a comprehensive mechanistic study to understand the  inactivation mechanism of (S)-3-Amino-(S)-4-fluorocyclopentenecarboxylic acid in hOAT. With the experimental and computational results, the compounds are roved to exhibits unusual structural conversions catalyzed by specific residues within hOAT, which leading to an enamine mechanism-based inactivation of hOAT through noncovalent interactions and covalent modification.

The research work carried out by the authors are appreciated.

The following are the minor comments to be addressed by the authors before the manuscript could be accepted in the Journal Molecules for publication.

1.       Rewrite the abstract with more significant findings and importance of the study.

2.       The materials method sections need to be improved with clear methodologies and strategies in collect the data.

3.       Introduction must be improved with motivation behind the study and recent relevant references.

4.       The crystal structure data should be provided with clear structural description.

5.         The DFT calculation methodologies and the result analysis part must be improved.

6.       The docking studies section needs to be reframed with clear discussion of methodology and significance of the obtained results.

7.       Provide the crystal structure details with CCDC number.

8.       I think in the conclusion section the reference is not required. It can be in results and discussion section.

With the above comments, I recommend for acceptance of the manuscript for publication after revision.

Author Response

Reviewer 1

The authors have carried out a comprehensive mechanistic study to understand the  inactivation mechanism of (S)-3-Amino-(S)-4-fluorocyclopentenecarboxylic acid in hOAT. With the experimental and computational results, the compounds are roved to exhibits unusual structural conversions catalyzed by specific residues within hOAT, which leading to an enamine mechanism-based inactivation of hOAT through noncovalent interactions and covalent modification.

The research work carried out by the authors are appreciated.

The following are the minor comments to be addressed by the authors before the manuscript could be accepted in the Journal Molecules for publication.

  1. Rewrite the abstract with more significant findings and importance of the study.

Response: The importance of the study was added as well as additional significant findings.

  1. The materials method sections need to be improved with clear methodologies and strategies in collecting the data.

Response: The biochemical and biological experimental procedures we implemented in this work were well characterized and published in our previous efforts. All the methodologies and strategies we used have been appropriately cited. Please see the summary as follows.

3.2. Expression and Purification of hOAT: Biochemistry 2017, 56 (37), 4951-4961. DOI: 10.1021/acs.biochem.7b00499. (Ref 17)

3.3. Aminotransferases and Coenzymes for Kinetic Studies: J Am Chem Soc 2015, 137 (7), 2628-2640. DOI: 10.1021/ja512299n (Ref 32); Anal Biochem 2013, 440 (2), 145-149. DOI: 10.1016/j.ab.2013.05.025 (Ref 33); ACS Med Chem Lett 2020, 11 (10), 1949-1955. DOI: 10.1021/acsmedchemlett.9b00672 (Ref13).

3.4. Denaturing Intact Protein and Small Molecule Mass Spectrometry: J Am Chem Soc 2020, 142 (10), 4892-4903. DOI: 10.1021/jacs.0c00193 (Ref14); J Am Chem Soc 2021, 143 (23), 8689-8703. DOI: 10.1021/jacs.1c02456 (Ref 25); J Am Chem Soc 2022, 144 (12), 5629-5642. DOI: 10.1021/jacs.2c00924 (Ref 44).

3.5. Theoretical pKa Calculations: J Am Chem Soc 2021, 143 (23), 8689-8703. DOI: 10.1021/jacs.1c02456 (Ref 25).

3.6. Electrostatic Potential (ESP) Charge Calculation: J Am Chem Soc 2021, 143 (23), 8689-8703. DOI: 10.1021/jacs.1c02456 (Ref 25).

3.7. Co-crystallization of hOAT with Compound 5: J Am Chem Soc 2020, 142 (10), 4892-4903. DOI: 10.1021/jacs.0c00193 (Ref14); J Am Chem Soc 2021, 143 (23), 8689-8703. DOI: 10.1021/jacs.1c02456 (Ref 25); J Am Chem Soc 2022, 144 (12), 5629-5642. DOI: 10.1021/jacs.2c00924 (Ref 44).

3.8. Transient State Methods: J Am Chem Soc 2021, 143 (23), 8689-8703. DOI: 10.1021/jacs.1c02456 (Ref 25).

3.9. Gibb’s Free Energy Calculation: J Am Chem Soc 2021, 143 (23), 8689-8703. DOI: 10.1021/jacs.1c02456 (Ref 25).

3.10. Molecular Docking Protocol: J Am Chem Soc 2020, 142 (10), 4892-4903. DOI: 10.1021/jacs.0c00193 (Ref14); ACS Med Chem Lett 2020, 11 (10), 1949-1955. DOI: 10.1021/acsmedchemlett.9b00672 (Ref13).

  1. Introduction must be improved with motivation behind the study and recent relevant references.

      Response: The introduction has been improved by additional comments regarding the motivation behind the study. Also, we have narrated the historical development process of compound 5 from the first FDA-approved inactivator vigabatrin and its monofluoride derivative AFPA to the cyclopentane analog FCP (4) with proper citations. Moreover, based on our prior experience with other similar cyclic analogues (J Am Chem Soc 2020, 142 (10), 4892-4903. DOI: 10.1021/jacs.0c00193 (Ref14); J Am Chem Soc 2021, 143 (23), 8689-8703. DOI: 10.1021/jacs.1c02456 (Ref 25);), we understand that minor structural modifications might lead to significant mechanical changes. In this work, we examined the inactivation mechanism of cyclopentene analogue 5, which exhibits much more potency than its parent cyclopentane compound 4 (ACS Med Chem Lett 2020, 11 (10), 1949-1955. DOI: 10.1021/acsmedchemlett.9b00672 (Ref13)). Utilizing a variety of approaches, we demonstrated that compound 5 inactivates hOAT through a distinct and novel mechanism compared to compound 4. These statements have all been made in the Introduction with appropriate references.

  1. The crystal structure data should be provided with a clear structural description.

      Response: A more detailed structural description was added to the co-crystallization section. Figure S2 and Figure S3 were added to the SI for better understanding.

  1. The DFT calculation methodologies and the result analysis part must be improved.

      Response: In the past, we applied the same DFT calculation methodology for prediction and published the pKa calculation results for other analogs (J Am Chem Soc 2020, 142 (10), 4892-4903. DOI: 10.1021/jacs.0c00193 (Ref14); J Am Chem Soc 2021, 143 (23), 8689-8703. DOI: 10.1021/jacs.1c02456 (Ref 25)). Some brief description with references was added to the main text and experimental section for better clarity.

  1. The docking studies section needs to be reframed with clear discussion of methodology and significance of the obtained results.

      Response: In the past, we applied the same molecular docking methodology and published results for other analogues (ACS Med Chem Lett 2020, 11 (10), 1949-1955. DOI: 10.1021/acsmedchemlett.9b00672 (Ref13); J Am Chem Soc 2020, 142 (10), 4892-4903. DOI: 10.1021/jacs.0c00193 (Ref14)). Similar docking poses were observed in the current work. Some description was added to the experimental section with references for better clarity.

  1. Provide the crystal structure details with CCDC number.

      Response: The crystal structure of the protein-ligand complex was submitted to the RCSB protein data bank (RCSB PDB). The PDB ID is 8EZ1. CCDC deposit is required for small molecule crystallography results, which were not presented by this manuscript, not for protein crystallography.

  1. I think in the conclusion section the reference is not required. It can be in results and discussion section.

      Response: We think the citations might be helpful to navigate the previous findings related to the current work. We’d like to keep these references.

With the above comments, I recommend for acceptance of the manuscript for publication after revision.

Reviewer 2 Report

The present manuscript represents a new contribution from the RB Silverman laboratory aiming to shed lights on the inactivation mechanism of hOAT by (S)-3-Amino-(S)-4-fluorocyclopentenecarboxylic acid (compound 5). Interestingly, this compound was shown to be a potent inactivator of  hGABA-AT, proceeding by an enamine mechanism, which exhibits 10-fold less inactivation efficiency (kinact/KI) against hOAT.

hOAT has recently emerged as a potential target for developing effective treatments for different types of cancers, namely in liver and in lung.

Compound 5 exhibits unusual structural conversions catalyzed by specific residues within hOAT, ultimately leading to an enamine mechanism-based inactivation of hOAT through noncovalent interactions and covalent modification.

The manuscript is very well written, clear, precise, and easy to understand. Complementary structural and mechanistic enzymology experiments (X-ray crystallography, denaturing intact protein-ligand MS, UV stopped-flow UV spectrophotometry (at 350, 420 and 580nm) - transient state kinetics and computational chemistry) have been well-designed and well-conducted.

Overall, the manuscript deserves to be published in Molecules.

The authors, however before acceptance should fully address the following issues that mainly concern the X-ray crystallography part:

I) In order to avoid the likely model bias, instead of the reported "omit" (Fo – Fc) maps, the authors should calculate instead "composite - omit" maps or even better Feature-Enhanced Maps,  modified 2mFobs - DFmodel sA weighted maps, (see Acta Crystallographica (2015) D71, 646).

II) The authors state: "The unique binding orientation of 5 also led to a water-mediated hydrogen bond interaction between the imine group of the ligand and Tyr55 residue".

Is this water molecule "structurally" conserved among the available hOAT crystal structures deposited in the PDB?

Would it affect the protonation state (pKa) of the neighboring titratable groups?

III) In Table S1, a value of 46,5% for CC1/2 is reported for the highest resolution shell. This is a rather  low value and it clearly indicates that the reported resolution of 1.91Å is over-estimated. Therefore the crystal structure should be re-refined at a lower resolution (the resulting CC1/2 value should  at least result of 95,0% or higher).

IV)  In Table S1, the average B (Å2) should be reported not only for the protein atoms ( it would be better to report the average B (Å2) for the protein atoms belonging to each of the 3 molecules in the asymmetric unit) but also for the atoms belonging to the three ligands (compound 5)  and the water molecules.

v) Which is the occupancy factor of each of the three ligands?

VI) The authors are kindly request to share with the reviewer the Validation report produced for the PDB ID 8EZ1 by the wwPDB validation service.

VII) The temperature used during the setup of the crystallization experiments should be reported.

Curiosity driven question:

"The freshly prepared enzyme was dialyzed with 50 mM potassium 613 pyrophosphate which contained 5 mM α-ketoglutarate at pH 8.0 overnight. The enzyme was then incubated with excess amount of compound 5 at 4 °C overnight. The pre-inactivated hOAT sample was buffer exchanged into 50 mM Tricine pH 7.8 and concentrated to a protein concentration of 6 mg/mL. Previously reported crystallization conditions27 were optimized using the hanging drop vapor diffusion method by varying PEG 6000 (8-12%), NaCl (100-250 mM), glycerol (0%-10%) with 100 mM Tricine pH 7.8 as the buffer"

Would the pH/ionic strength affect the covalent / noncovalent protein-ligand ratio?

Minor issue.

Please spell in full (when first appear in the text) the CCD and SVD abbreviations i.e. "Charge-Coupled Device" and "Singular Value Decomposition".

Author Response

Reviewer 2

The present manuscript represents a new contribution from the RB Silverman laboratory aiming to shed lights on the inactivation mechanism of hOAT by (S)-3-Amino-(S)-4-fluorocyclopentenecarboxylic acid (compound 5). Interestingly, this compound was shown to be a potent inactivator of  hGABA-AT, proceeding by an enamine mechanism, which exhibits 10-fold less inactivation efficiency (kinact/KI) against hOAT.

hOAT has recently emerged as a potential target for developing effective treatments for different types of cancers, namely in liver and in lung.

Compound 5 exhibits unusual structural conversions catalyzed by specific residues within hOAT, ultimately leading to an enamine mechanism-based inactivation of hOAT through noncovalent interactions and covalent modification.

The manuscript is very well written, clear, precise, and easy to understand. Complementary structural and mechanistic enzymology experiments (X-ray crystallography, denaturing intact protein-ligand MS, UV stopped-flow UV spectrophotometry (at 350, 420 and 580nm) – transient state kinetics and computational chemistry) have been well-designed and well-conducted.

Overall, the manuscript deserves to be published in Molecules.

The authors, however before acceptance should fully address the following issues that mainly concern the X-ray crystallography part:

  1. I) In order to avoid the likely model bias, instead of the reported omit” (Fo – Fc) maps, the authors should calculate instead "composite - omit"” maps or even better Feature-Enhanced Maps,  modified 2mFobs– DFmodelsA weighted maps, (see Acta Crystallographica (2015) D71, 646).

Response: Although the provided omit map is considered a type of unbiased validation map via omitting the built ligand, the authors accept the reviewer’s suggestion. The composite omit maps have been used in the revised Figure 5.

  1. II) The authors state: "The unique binding orientation of 5also led to a water-mediated hydrogen bond interaction between the imine group of the ligand and Tyr55 residue".

Is this water molecule "structurally" conserved among the available hOAT crystal structures deposited in the PDB?

Would it affect the protonation state (pKa) of the neighboring titratable groups?

Response: The observed water molecule in proximity to Tyr55 is present in the structure of the holoenzyme (PDB ID: 1OAT). This water molecule could shield Tyr55 from a potential direct interaction with Arg180 which was previously observed. At the pH of 7.8, this water molecule is hydrogen-bonded to Tyr55 unless it gets pushed away by the ligand. For example, in the co-crystal structure of hOAT with (S)-3-amino-4,4-difluorocyclopent-1-enecarboxylic acid (SS-1-148) (PDB ID: 7LK0) Tyr55 forms hydrogen bonds with the carboxylate group of the ligand, leaving no space available for the water molecule.

III) In Table S1, a value of 46,5% for CC1/2 is reported for the highest resolution shell. This is a rather low value and it clearly indicates that the reported resolution of 1.91Å is over-estimated. Therefore the crystal structure should be re-refined at a lower resolution (the resulting CC1/2 value should at least result of 95,0% or higher).

Response: The authors believe that the inclusion of weak high-resolution data has the capacity to improve the quality of the model, despite the introduced noise (please see Current Opinion in Structural Biology (2015) Volume 34, Pages 60-68). Exclusion of the weak-signal data could underestimate the information on the crystal structure and, in turn, result in subsequent model errors. For this reason, the authors prefer to use CC1/2 (≥ 0.3) as a more robust criterion to determine the resolution cut-off. The authors believe that the field of protein crystallography has evolved to accept the new cut-off criteria using CC1/2 (≥ 0.3).  This trend is evidenced by major data processing software such as Autoproc and Xia2 now both using this criterion as the “default” setup in their most recent versions. 

  1. IV)  In Table S1, the average B (Å2) should be reported not only for the protein atoms ( it would be better to report the average B (Å2) for the protein atoms belonging to each of the 3 molecules in the asymmetric unit) but also for the atoms belonging to the three ligands (compound 5)  and the water molecules.

Response: B factors for all three ligands molecules were added to the statistics table.

  1. v) Which is the occupancy factor of each of the three ligands?

Response: The occupancy factor for all three copies of the ligand is equal to 1.0 (100% occupancy). This was added to the manuscript.

  1. VI) The authors are kindly requested to share with the reviewer the Validation report produced for the PDB ID 8EZ1 by the wwPDB validation service.

Response: The authors are glad to share the PDB Validation report. Please, see attached.

VII) The temperature used during the setup of the crystallization experiments should be reported.

Response: Information about the temperature was added to the methodology part of the manuscript.

Curiosity driven question:

"The freshly prepared enzyme was dialyzed with 50 mM potassium 613 pyrophosphate which contained 5 mM α-ketoglutarate at pH 8.0 overnight. The enzyme was then incubated with excess amount of compound 5 at 4 °C overnight. The pre-inactivated hOAT sample was buffer exchanged into 50 mM Tricine pH 7.8 and concentrated to a protein concentration of 6 mg/mL. Previously reported crystallization conditions27 were optimized using the hanging drop vapor diffusion method by varying PEG 6000 (8-12%), NaCl (100-250 mM), glycerol (0%-10%) with 100 mM Tricine pH 7.8 as the buffer"

Would the pH/ionic strength affect the covalent / noncovalent protein-ligand ratio?

Response: That is a great question. A pH study of hOAT and its native and alternative substrates has been conducted before (please see J. Biol. Chem. (2022), 298(6), 101969). In that study, it was clearly shown that lower pH environment severely reduces the kinetics of the first-half reaction, but no changes in the final product spectra were observed. However, that study did not account for hOAT inhibitors, which leaves the question of the equilibrium of final adducts open.

Minor issue.

Please spell in full (when first appear in the text) the CCD and SVD abbreviations i.e. "Charge-Coupled Device" and "Singular Value Decomposition".

Response: The full names were added to the main test.